# Germline mutation landscape of DNA damage repair genes in African Americans with prostate cancer highlights potentially targetable *RAD* genes

Indu Kohaar [1,2,8 ✉], Xijun Zhang[3,4,8], Shyh-Han Tan [1,2], Darryl Nousome[1,2], Kevin Babcock[1], Lakshmi Ravindranath[1,2], Gauthaman Sukumar[3,4], Elisa Mcgrath-Martinez[3,4], John Rosenberger [3,4], Camille Alba [3,4], Amina Ali[1,2,5], Denise Young[1,2], Yongmei Chen[1], Jennifer Cullen[1], Inger L. Rosner[1], Isabell A. Sesterhenn[6], Albert Dobi[1,2], Gregory Chesnut[1,5], Clesson Turner[3], Clifton Dalgard[3,4], Matthew D. Wilkerson[3,4], Harvey B. Pollard[3,4], Shiv Srivastava [1,7,8] & Gyorgy Petrovics [1,2,8 ✉]

In prostate cancer, emerging data highlight the role of DNA damage repair genes (DDRGs) in aggressive forms of the disease. However, DDRG mutations in African American men are not yet fully defined. Here, we profile germline mutations in all known DDRGs ($N = 276$) using whole genome sequences from blood DNA of a matched cohort of patients with primary prostate cancer comprising of 300 African American and 300 European Ancestry prostate cancer patients, to determine whether the mutation status can enhance patient stratification for specific targeted therapies. Here, we show that only 13 of the 46 DDRGs identified with pathogenic/likely pathogenic mutations are present in both African American and European ancestry patients. Importantly, *RAD* family genes (*RAD51, RAD54L, RAD54B*), which are potentially targetable, as well as *PMS2* and *BRCA1*, are among the most frequently mutated DDRGs in African American, but not in European Ancestry patients.

[1] Center for Prostate Disease Research, John P. Murtha Cancer Center Research Program, Department of Surgery, Uniformed Services University of the Health Sciences and the Walter Reed National Military Medical Center, Bethesda, MD 20817, USA. [2] Henry M. Jackson Foundation for the Advancement of Military Medicine, Bethesda, MD 20817, USA. [3] The American Genome Center, Precision Medicine Initiative for Military Medical Education and Research (PRIMER), Uniformed Services University of the Health Sciences, Bethesda, MD 20814, USA. [4] Department of Anatomy, Physiology and Genetics, Uniformed Services University of the Health Sciences, Bethesda, MD 20814, USA. [5] Urology Service, Walter Reed National Military Medical Center, Bethesda, MD 20814, USA. [6] Joint Pathology Center, Silver Spring, MD 20906, USA. [7] Present address: Department of Biochemistry and Molecular & Cell biology, Georgetown University School of Medicine, Washington, DC 20057, USA. [8] These authors contributed equally: Indu Kohaar, Xijun Zhang, Shiv Srivastava, Gyorgy Petrovics. ✉email: ikohaar@cpdr.org; gpetrovics@cpdr.org

The comprehensive knowledge of germline and somatic mutations in prostate cancer (CaP) has started to propel precision medicine opportunities. Genetic testing for inherited mutations has been rapidly increasing in patients with advanced CaP[1–4]. DNA damage repair genes (DDRGs) play a critical role in genomic stability, homologous recombination (HR) and mismatch repair and defects in these functions are central to diverse cancers. Genomic changes in DDR pathways are more prevalent in metastatic CaP (mCaP) and in castration resistant CaP. Therefore, DDRG mutations at these advanced stages of CaP are increasingly considered for patient selection for targeted therapies. Along these lines, Poly ADP Ribose Polymerase (PARP) inhibitors have been shown to extend overall survival in cancer patients with targetable DDRG mutations[4,5], while germline mutations in DNA mismatch repair (MMR) pathway help in selecting patients for immunotherapy[6]. In addition, patients with DDRG mutations were reported to respond better to hormone or chemotherapy in mCRPC[7,8].

While the majority of CaP research to identify P/LP germline or somatic mutations has been performed in cohorts of European descent, there were indications of ancestral differences in frequencies and distribution of germline mutations[9,10]. Thus, it is critical to extend similar studies to individuals from diverse ancestries. The Philadelphia Prostate Cancer Consensus Report[2] made a strong recommendation for the urgent need of research in DDRGs of African American (AA) CaP patients.

We and others have noticed that commercial DDRG mutation tests were generated from data based on relatively small subsets of known DDRGs and using patient cohorts over-represented by European Ancestry (EA) CaP patients at the metastatic stage[11–13]. It is widely recognized that AA men are disparately affected by CaP as reflected in higher incidence, earlier onset, more aggressive progression, and higher disease specific death[14–16]. While post-treatment outcome related disparities for AA CaP are significantly reduced in post-PSA testing era, especially in equal access healthcare systems such as in US military, we have continued to discover highly significant molecular genetic differences between EA and AA CaP[9,17,18].

In the present study, we have profiled all known DDRGs in an unbiased way using germline DNA of a cohort of 600 patients, drawn equally from EA and AA men matched for age and stage of the disease. Genomic DNA specimens were derived from blood samples collected from active duty and retired military personnel treated with radical prostatectomy (RP) at the Walter Reed National Military Medical Center (WRNMMC) and archived at the Center for Prostate Disease Research (CPDR) biospecimen bank[19]. In this work, we have developed foundational data by evaluating the complete spectrum and prevalence of germline mutations in all known DDRGs in both AA and EA CaP patients who had equal access to screening, treatment, and follow-up. These data may aid in future goals of personalized medicine by enhancing the stratification of patients for targeted therapeutic options and by providing genetic counseling, specifically to high-risk families.

## Results

**Patient characteristics and ancestry confirmation.** In the present study, a total of 600 archived blood genomic DNA specimens derived from RP patients were profiled for all DDRG mutations. The final analysis was performed in 531 CaP patients, including 259 AA and 272 EA men, after excluding 11.5% (69/600) of the patients because of low quality sequencing results due to DNA yields, fragment size and contamination ($N = 36$), or mismatch between genomic ancestry and self-reported race ($N = 33$)

(Fig. 1). The patient characteristics are summarized in Table 1. No significant difference was found for any of the clinical variables between AA and EA men. To further determine population similarities between our cohort and reference control cohorts with common variants, we compared common variant allele frequencies (>1%) and observed a high degree of correlation between AA prostate cancer and AA controls[20], as well as between EA prostate cancer and EA control individuals[20] ($r^2 > 0.99$), demonstrating that population structure was similar across cases and controls in each ancestry group (Supplementary Fig. 1).

**Ancestral differences in DDRG germline mutations.** We estimated the allele frequencies in this cohort (259 AA and 272 EA men) for all variants in 276 DDRGs by performing comparative analyses with a control reference cohort[20]. A total of 6293 non-silent and splicing variants in the DDRG region in this cohort were analyzed. 17.5% (1103/6293) of the variants were found as novel variants, as these are not present in control database. Considering Pathogenic/ Likely Pathogenic (P/LP) variants (based on ClinVar and InterVar program), a total of 62 and 76 variants in AA and EA men, respectively, were identified as germline mutations, including several known and novel germline mutations[21]. Overall, 23.5% (125/531) patients had a germline mutation in at least one of the 276 known DDRGs. Compared to relevant literature, this study revealed germline mutation rate of 22.8% in AA men and 24.3% in EA men, which is comparatively higher, especially in AA men, than reported by Nicolosi et al.[13] (AA, 10.1%; CA 17.8%) and Sartor et al.[12] (AA, 7.5%; EA, 13.9%), although the cited studies were based on targeted sequencing approach. Out of 276 DDRGs, 46 genes had germline mutations, and only 28.3% (13 of 46) were common among AA and EA patients, underscoring clear ancestral differences in the distribution of DDRG germline mutations (Fig. 2). Closer evaluation of the ancestral distribution of germline mutations identified that mutations in *RAD51* and *PMS2* genes were enriched in AA compared to EA CaP patients, with *p* values of 0.0621 and 0.0268, respectively. On the other hand, *FANCA* was significantly more frequently mutated in EA men compared to AA patients ($P = 0.0076$) (Fig. 2).

We also performed gene-based total variant frequency test for DDR genes with at least three carriers, in CPDR cohort of AA or EA men, in comparison to corresponding control cohorts[22]. *BRCA1, RAD51, FANCL, POLG, PMS2,* and *RAD54L* were significantly altered in AA men ($P < 0.05$), while germline mutations in *POLG1* and *OGG1* were significantly enriched in men with EA ($p < 0.05$) (Fig. 3). A review of literature (Supplementary Data 1) on similar studies has shown that the present study is the most comprehensive study with the largest number of DDRGs ($N = 276$) and germline mutations ($N = 46$) included in both AA and EA men. The most frequent (over 1% carrier frequency) and potentially clinically targetable mutations[11,23], including the novel ones in AA patients, were confirmed by the WGS-independent ddPCR method (Supplementary Data 2). Remarkably, ddPCR assay results agreed with WGS results in 99.15% (117 of 118) of the cases (Supplementary Fig. 2). In addition to using the ClinVar and InterVar program, we also functionally scored and validated the germline mutations using an enGenome in silico tool based on ACMG, AMP, ClinGen guidelines[24–26] (Supplementary Data 3).

**Higher frequency of germline mutations in *RAD* genes may benefit AA patients with targeted therapy options.** Considering clinical utility, the most important subset of DDRGs are the ones which harbors targetable mutations. A major finding of this study

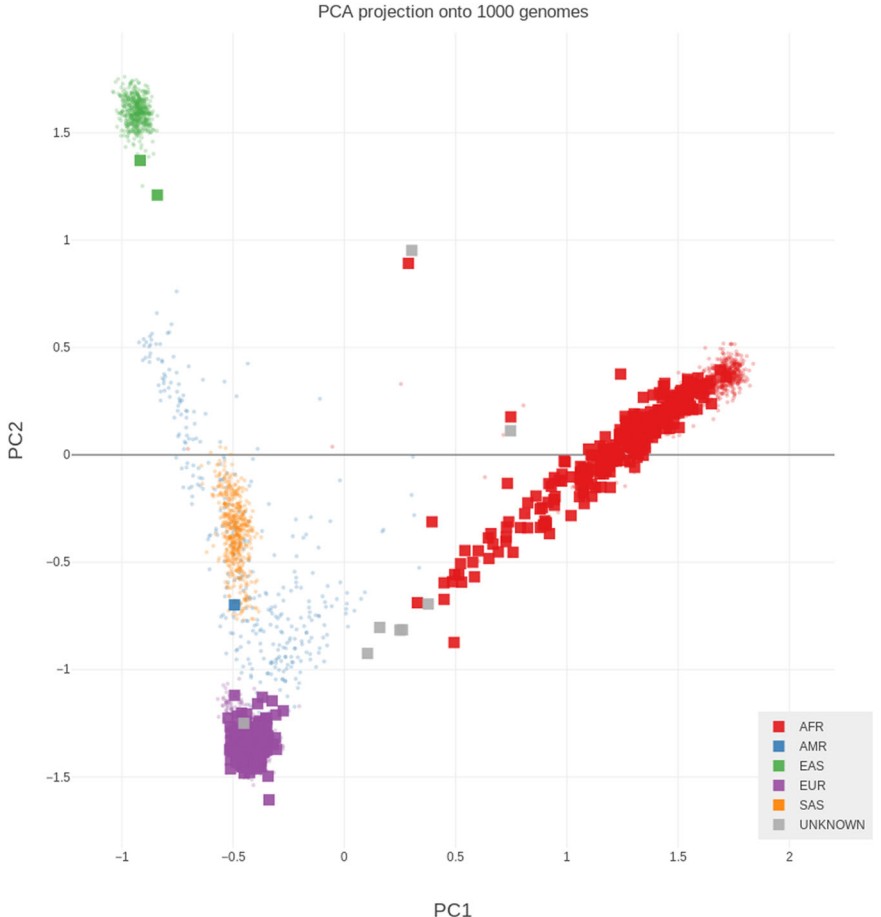

**Fig. 1 Principle Component Analysis (PCA) of Ancestry.** Using Peddy program[48], a PCA analysis was conducted on each patient in this study (large square points). Ancestry predictions were made based on an SVM (support vector machine) model trained on the 1000 Genome samples[49] ($n = 2504$, small background points). Five superpopulations under 1000 G project are: Africans (AFR) with seven populations, Americans (AMR) with four populations; East Asians (EAS) with five populations; Europeans (EUR) with five populations and South Asians (SAS) with five populations. AFR includes ASW population (African Ancestry in Southwest USA). Source data are provided as a Source Data file.

is that several *RAD* family genes (*RAD51, RAD54L, RAD54B),* as well as *PMS2* and *BRCA1*, were among the most frequently mutated DDRGs in AA patients, but not in EA patients, when compared to the relevant control datasets (Fig. 3). These genes are part of targetable DDRG pathways[11,23,27], specifically, the HR[28–32] and mismatch repair (MMR) pathways[6,10,33,34], suggesting potential benefit for AA men. Patients with DDRG mutation in the HR pathway may respond well to PARP inhibitor therapy, and patients with DDRG mutation in the MMR pathway may respond to immune checkpoint inhibitor therapy. *RAD51* encodes a key enzyme in HR pathway which is a therapeutic target in oncology[35] and several RAD51 inhibitors have been previously reported[36]. In addition, *RAD54B* and its functionally related paralog *RAD54L* are now being explored as precision medicine targets, considering their key role in maintaining and repairing genome stability after DNA damage[37]. Because *RAD* genes are functionally related, we grouped all *RAD* mutations together and then observed a greater mutation rate in AA (6.95%) than in EA patients (1.10%) ($P = 0.001$) (Supplementary Table 1). Pathway analysis using 14 published DDR pathways[38] showed that the HR pathway (with 15 mutations including those in *BRCA1, BRCA2, RAD51* in AA and *BLM, NBN* in EA men) is associated with faster disease progression to BCR ($p$ 0.02; HR 3.34) (Figs. 4, 5), while the NHEJ pathway (with three mutations) is marginally associated with metastasis ($p$ 0.045) (Supplementary Table 2).

**DDRG mutation panel: potential clinical utility in AA and EA CaP patients.** Sixteen of the 46 DDRGs with germline mutations in this cohort were in a potentially targetable DDR pathway (HR and MMR). Based on our WGS and ddPCR results, screening for germline mutations by a test panel that comprise these 16 genes would detect 57 of 531 patients (10.7%), including 35 AA and 22 EA men. A test based on a panel of 8 of the 16 potentially targetable DDRGs, which have over 1% germline mutation frequency, would detect 41 of 531 cases (7.7%). Interestingly, a test using this eight-gene panel (*FANCA, MSH6, FANCL, RAD54B, BRCA1, PMS2, RAD54L, RAD51*), selected based on higher frequency and clinical relevance[27], would detect 11.6% (30 of 259) of AA CaP patients and only 5.8% (16 of 272) of EA CaP patients with potentially targetable mutations ($P = 0.021$; Supplementary Table 3).

**DDRG germline mutations in AA men associate with poor disease outcome.** In this patient cohort with up to 25 years of clinical follow-up, germline mutations in any of the DDRGs was associated with shorter time to biochemical recurrence (BCR) (Kaplan–Meier analysis, $P = 0.044$) in AA patients, but not in EA patients ($P = 0.74$) (Fig. 5). In another analysis, an almost twofold higher percentage of BCR was found among AA patients with DDRG germline mutations (23.1%), compared to patients with no mutations (11.4%; $P = 0.032$). However, in men with EA this difference was not significant (12.7% of patients who had

**Table 1 Clinico-pathologic characteristics and DDRG mutation status of 531 patients with prostate cancer stratified by ancestry.**

| | African American (N = 259) | European ancestry (N = 272) | p value |
|---|---|---|---|
| Gleason Grade | | | 0.166 |
| 3 + 3 | 127 (51.6%) | 160 (59.9%) | |
| 3 + 4 | 69 (28.0%) | 63 (23.6%) | |
| 4 + 3/8-10 | 50 (20.3%) | 44 (16.5%) | |
| Missing Data | 13 | 5 | |
| Diagnosis Age (in years) | | | 0.078 |
| Old (>55 years) | 193 (74.5%) | 220 (80.9%) | |
| Young (≤55 years) | 66 (25.5%) | 52 (19.1%) | |
| PSA Category (ng/ml) | | | 0.241 |
| (<4) | 59 (23.0%) | 81 (29.8%) | |
| (4-9) | 164 (63.8%) | 165 (60.7%) | |
| (10-20) | 26 (10.1%) | 19 (7.0%) | |
| (>20) | 8 (3.1%) | 7 (2.6%) | |
| Missing Data | 2 | 0 | |
| Pathological T Stage | | | 0.041 |
| T1a-T2a | 6 (2.6%) | 17 (6.7%) | |
| T2b-T2c | 166 (72.2%) 158 | 158 (62.2%) | |
| T3a-T3c | 58 (25.2%) | 78 (30.7%) | |
| T4 | 0 (0.0%) | 2 (0.8%) | |
| Missing Data | 29 | 18 | |
| BCR | | | 0.788 |
| No | 203 (86.0%) | 224 (85.2%) | |
| Yes | 33 (14.0%) | 39 (14.8%) | |
| Missing Data | 23 | 9 | |
| Metastasis | | | 0.451 |
| No | 250 (96.5%) | 259 (95.2%) | |
| Yes | 9 (3.5%) | 13 (4.8%) | |
| Any DDRG Mutations | | | 0.687 |
| No Mutation | 200 (77.2%) | 206 (75.7%) | |
| Any Mutation | 59 (22.8%) | 66 (24.3%) | |
| Number of DDRG Mutations Per Patient | | | 0.170 |
| 0 | 200 (77.2%) | 206 (75.7%) | |
| 1 | 56 (21.6%) | 56 (20.6%) | |
| 2 | 3 (1.2%) | 10 (3.7%) | |

Two-sided test was performed to assess the p value; P values < 0.05 is considered significant.
PSA Prostate Serum Antigen, BCR Biochemical Recurrence, DDRG DNA Damage Repair Gene.

BRCA1/2 were more frequent in AA than in EA CaP patients[10,40]. Unexpectedly, there was a higher percentage of AA men (P = 0.021; OR = 2.10), compared to men with EA, with potentially targetable DDRG germline mutations[4,6,11,23,28–31]. When all RAD germline mutations are combined, a significantly higher (P = 0.001) mutation rate was found AA than in EA patients. In addition, in AA men, BRCA1, RAD51, FANCL, PMS2 and RAD54L (p value < 0.05), and in EA men, OGG1[41] (p value < 0.05) was also significantly altered when compared to controls. Interestingly, POLG[42,43] was significantly altered in both AA and EA prostate cancer cases compared to controls. This is similar to what was recently reported by Wu et al.[44] in CaP of a Chinese population, thereby, implying role of POLG as a potential CaP pre-disposition gene in different populations. AA patients with germline mutation in DDRGs showed association with poor disease outcome/progression (BCR and time to BCR). These findings underline ancestral differences and point to the utility of a new generation of genetic tests inclusive of both AA and EA men. We identified 16 DDRGs as potentially targetable in the HR and MMR pathways[4,11,28], and propose that they should be considered for the selection of patients for early targeted therapy (e.g., by PARP inhibitors or immune checkpoint inhibitors). In addition, the percentage of patients with all DDRG germline mutations is unexpectedly robust (23%) and should be considered for early genetic testing and genetic counseling both in AA and in EA patients and family members.

A comprehensive study on DDRG germline mutations in CaP with a significant subset of AA patients was published by Nicolosi et al.[13], which identified eight DDRGs in the AA cohort (N = 227) with germline mutation. In our AA cohort (N = 259), 23 DDRGs had germline mutation (including seven of the eight genes reported in the Nicolosi et al. study). In another study by Sartor et al.[12], 7.5% (16/214) AA men had pathogenic germline mutations in seven genes, of which six were identified in our analysis. Ledet et al.[40], in a recent study on germline mutations in mCaP, reported that 6% (11/188) of the AA patients harbored P/LP while 55% (104/188) had variants of unknown significance, using commercially available gene panel. In addition, they found that AA were more likely to have a P/LP in BRCA1 compared to EA men. Similar to our study, Castro et al.[8] identified germline mutations in RAD54L but at a relatively lower (0.2%) frequency than ours (1.5% AA; 0.7% EA). Matejcic et al.[39] reported comparatively lower frequency of rare pathogenic variants (2.1% in controls; 3.6% in cases and 5.7% in metastatic patients) in a combined analysis of AA and Ugandan men in a case-control study based on 19 DDRGs. The study however, validated the findings for the association of DDRGs with aggressive disease in AA men. They found that highest risks for aggressive disease were observed with pathogenic variants in the ATM, BRCA2, PALB2, and NBN genes. In our analysis, we do find that BRCA2, NBN are 2 of 5 genes (RAD51, BRCA1, BRCA2, BLM, NBN), where germline mutations are enriched in HR pathway associating with BCR. However, several of the most frequently mutated genes in AA CaP in our study were not tested by these earlier studies[12,13,39,45]. The main reasons for nearly three times higher percentage of patients with DDRG germline mutations in our study as compared to other reports, including therapeutically targetable ones for AA men, are our large panel of 276 DDRGs and the use of WGS platform for unbiased evaluation of all DDRGs. Future replication studies in an independent similarly designed patient cohort are warranted to validate our findings and improve the significance of true associations.

Presently, baseline screening for DDRG mutations in the setting of primary CaP is not included in the CaP guidelines[46,47]. Testing is clinically indicated in one or more of the following scenarios: Individuals with any blood relative with a known

germline mutation had BCR, in comparison to 15.5% with no mutation, P = 0.59). A similar trend for higher frequency of DDRG mutations was observed for NCCN high-risk category (P = 0.021) and clinical T stage (P = 0.021) in AA men (Supplementary Tables 4, 5). The aggregation of germline mutations by pathway showed that HR genes have a significant risk of disease progression to BCR (P = 0.018) in our combined AA and EA patient cohort.

## Discussion

Comprehensive analysis of all 276 DDRGs revealed that 46 of these genes had germline mutations in this cohort (N = 531). Of these, 23 DDRGs were mutated in AA patients (N = 259), which is about three times more than what was reported in similar recent studies[12,13,39,40]. In line with our hypothesis, the specific DDRGs with mutations were surprisingly different between AA and EA patients, only 13 of the 46 were common. Germline mutations in RAD51, RAD54L, RAD54B, PMS2, and BRCA1 were enriched in AA as compared to EA men while germline mutations in FANCA were present only in EA men. This is consistent with our previous study where we found that germline variants in

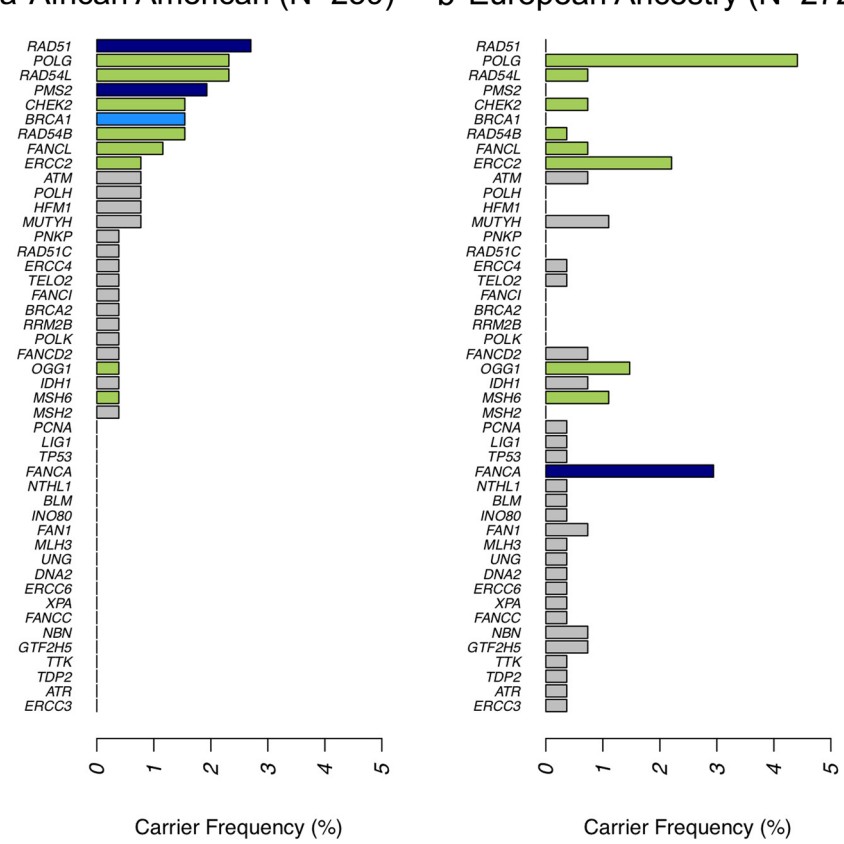

**Fig. 2 Germline Mutation Carrier Rate in Prostate Cancer Cases.** Carrier frequencies were calculated as the proportion of patients carrying at least one rare germline mutation on the DNA Damage Response Gene (DDRG). **a** Germline mutation carrier frequency in mutated DDR genes in African American (AA) cohort. **b** Germline mutation carrier frequency in mutated DDR genes in cohort of men with European Ancestry (EA). DDR genes with at least three carriers in CPDR cohort of AA or EA men are tested. A two-sided Fisher Exact Test was performed to assess the $p$ values. Colors of the bars indicate Fisher Exact Test $P$ values of AA vs. EA comparison. Dark blue: $P < 0.05$; Light blue: $0.05 < P < 0.1$; Green: $P > 0.1$; Gray: not tested. Germline mutations in *RAD51* and *PMS2* genes are enriched in African American cohort as compared to EA men with $P$ values of 0.0063 and 0.0271 respectively. Germline mutations in *FANCA* gene is enriched in EA men (Fisher $P$ value is 0.0075). Genes were sorted by carrier rate in AA cohort. Source data are provided as a Source Data file.

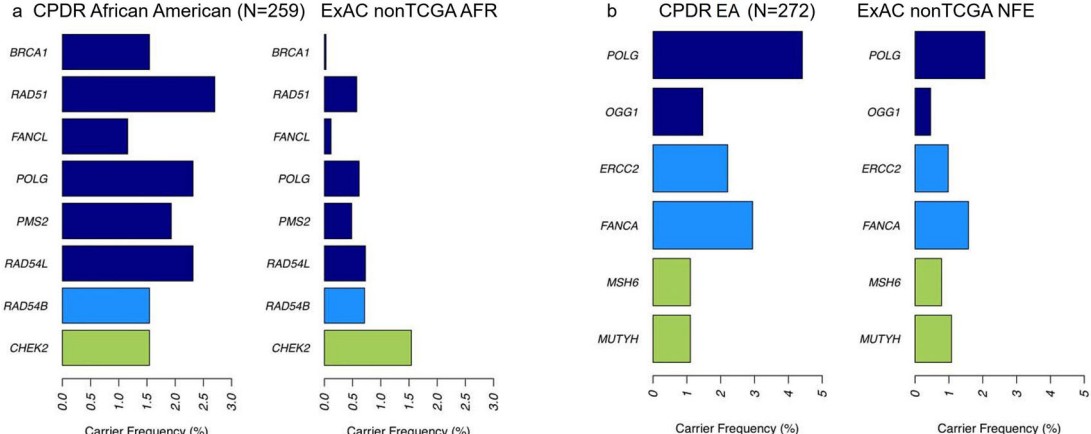

**Fig. 3 Gene Based Total Frequency of DDRG Germline Mutations Compared to Relevant Control Database.** Gene based total frequency tests were performed using germline mutants on DDR genes. Two-sided Fisher's exact tests were conducted using two by two table of number of carriers and non-carriers in this cohort and corresponding cohort in ExACnonTCGA. DDR genes with at least three carriers in CPDR AA or European Ancestry (EA) cohort are tested, (**a**) Bar plot of total frequency test results by comparing AA cohort and ExACnonTCGA AFR (**b**) Bar plot of total frequency test results by comparing cohort of EA men and ExACnonTCGA NFE. Colors of the bars indicate Fisher Exact Test $P$ values of case vs. control comparison. Dark blue: $P < 0.05$; Light blue: $0.05 < P < 0.1$; Green: $P > 0.1$. Source data are provided as a Source Data file.

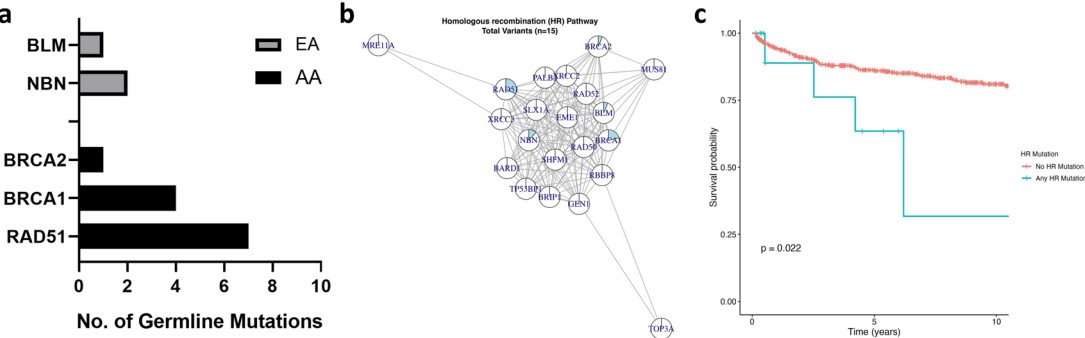

**Fig. 4 HR DDRG Pathway based Germline Mutations Profiling in AA and European Ancestry Men. a** Frequency of germline mutations in HR pathway in AA and EA men (**b**) The pie chart indicates the genes (*RAD51, BRCA1, BRCA2, BLM, NBN*) harboring 15 HR pathway mutations (**c**) Patients with variants in HR pathway had developed BCR faster than the patients without HR mutations, when longitudinally followed up after RP (*P* = 0.022, HR 3.34). The Kaplan–Meier curves were plotted, and log-rank tests were used to assess statistically significant differences between the two curves. Two-sided test was performed to assess the *p* value. Source data are provided as a Source Data file.

pathogenic/likely pathogenic (LP) variant in a cancer suscept-ibility gene; a person with personal history of CaP or family history of cancer or high-risk CaP or mCaP. The currently commercially available DDRG mutation assays rely on a relatively small number of genes which are not frequently mutated in our AA patient cohort. The present study offers ancestry specific significant findings that warrant future studies on profiling DDRGs in both early and advanced stages of CaP, especially in AA men. It may lead to new additions to the commercially available germline mutation test panels, especially for men of AA ancestry with CaP, who are under-represented in CaP genome studies and are disparately affected by this disease.

## Methods

**Patient cohort**. This retrospective cohort-based study involved 600 CaP patients (300 AA and 300 EA men), who were treated by RP at the WRNMMC. Only those patients were included who consented to enrollment in the CPDR biospecimen databank and multicenter national clinical database. The databases have been approved by the Institutional Review Boards (IRBs) at the WRNMMC and the Uniformed Services University of the Health Sciences (USUHS) in Bethesda, Maryland. All subjects provided written informed consent and agreed to provide blood specimens under our IRB approved protocols (WRNMMC-2017-0122, WRNMMC IRB # 393738 and WRNMMC # 385525). These patients, who are unselected for family history, are under the equal access DoD healthcare system. were followed up for up to 25 years, and provided extensive demographic, clinical, pathological, treatment, and outcome data. AA and EA patients were matched for age, pathological grade and stage (Table 1).

**Ancestry determination**. Sample ancestry were predicted using Peddy program[48], which used a support vector machine (SVM) model trained on individuals of diverse ancestries from the 1000 Genome Project reference panel[49]. Thirty-three samples with a predicted ancestry different than self-reported ancestry were excluded from further analysis.

**DNA extraction and whole genome sequencing library preparation**. Genomic DNA was extracted from peripheral blood lymphocytes of 600 patients using Qiagen DNeasy Blood kit at the CPDR. DNA concentration was normalized to 30 ng/ul using Qubit and a total of 3600 ng was used for WGS. 600 PCR-free libraries were prepared at The American Genome Center (TAGC) with 97.6% success rate. The quality of the sequencing libraries, including fragment size and concentration, were assessed before sequencing. Samples that passed quality control were sequenced on the NovaSeq system using a paired-end protocol (2× 150 bp, 400 bp insert size) with a minimum of 37× coverage.

**Sequencing quality assessment, data processing, cohort VCF file generation and annotation**. BCL files were converted to FASTQ files by bcl2fastq software. Paired-end reads were aligned to hg38 human reference genome using Isaac Aligner to generate BAM files. Using the resulting BAM files, single sample variant calls were made by Strelka2 Variant Caller[50]. Sequencing quality control was performed by Illumina Analysis Software and Picard (https://broadinstitute.github.io/picard/). Quality assessment metrics used in the process include mean coverage depth, total number of uniquely aligned reads, percent alignment, mean base quality. The WGS mean coverage was 37X. To evaluate within-sample contamination, we set ContEst

5% as the cutoff which excluded 17 samples from further analysis. Overall, sample-level quality control analysis identified 69 samples for filtering before subsequent study-level analyses.

To join variants from all samples, we used gvcfgenotyper (https://github.com/illumina/gvcfgenotyper) v. 2018.10.15) to merge sample genome VCF files into a cohort VCF file. Multi-allelic variants in the resulting cohort VCF file were split by bcftools into separate sites (https://github.com/samtools/bcftools v.1.9). Cohort level variants on autosomal chromosomes (chromosome 1-22) and chromosome X were further filtered by the following criteria: (1) the proportion of samples with non-reference alleles having a PASS filter (VQ, under FT tag) from individual sample genome VCF files is >90%; (2) The proportion of samples having a minimum genotype quality score of 20 (GQ ≥ 20) is >90%. To provide functional annotation, the filtered cohort VCF was annotated by ANNOVAR program[51].

**Interpretation of variants**. In this study, we focused on DDRGs. Using a published list of 276 DDRGs from Knijnenburg et al[38], we queried UCSC MySQL database hg38 refseq table and created a BED file based on genomic locations of start and stop sites of genes. Using this BED file, the filtered whole genome cohort VCF file was sliced to the DDRG regions. We applied InterVar program[52] to classify each variant on the DDRG regions. We took variants that were classified as Pathogenic (P) or Likely Pathogenic (LP) in either InterVar or ClinVar[53]. The variants were further functionally scored by enGenome in silico tool based on ACMG, AMP, ClinGen guidelines[24–26]. In addition, variants were further filtered by population allele frequency in gnomAD African/African_American (AFR) and non-Finnish European (NFE) populations. Specifically, variants having population allele frequencies >1% in either gnomAD_AFR or gnomAD_NFE were excluded from further analysis (http://gnomad.broadinstitute.org/)[54].

**ExAC nonTCGA data as healthy control**. Reference healthy control data were obtained from the publicly available Exome Aggregation Consortium (ExAC) database (http://exac.broadinstitute.org)[20]. ExAC non-TCGA dataset was selected as the control cohort for the following reasons: (1) ExAC is the largest whole exome database (N = 60,706); (2) ExAC database contains African/AA and NFE populations, which match the two ancestry groups in our study; (3) Individuals with no history of cancer are included in ExAcnonTCGA control dataset; (4) Information of all variants from this database is publicly available in VCF format. ExAC-nonTCAG VCF file was downloaded from ftp://ftp.broadinstitute.org/pub/ExAC_release/release0.3.1/subsets/ExAC_nonTCGA.r0.3.1.sites.vep.vcf.gz. To make this VCF file comparable with our data, we converted this file from hg19 to hg38 reference genome using UCSC Liftover tool. Using gene coordinates obtained from UCSC, we sliced ExAC VCF to DDRG regions. The resulting DDRG VCF file was further filtered by the following metrics: VQSLOS > −2.632 and Inbreeding Coeff > −0.8[54]. Using bcftools merge module, the filtered ExAC VCF was merged with the filtered DDRG VCF file of this study cohort.

**Population allele frequency comparison**. Allele frequencies of AA and EA patients were compared with ExAC African/African American (AFR) and NFE cohorts, and gnomAD African/AA and NFE cohorts, respectively. All PASS filter SNPs in DDRG region were used. Pearson's chi-square correlation analyses were performed using R.

**Gene based total carrier frequency test**. Gene based total frequency test[22] was performed where we combined the frequency of rare pathogenic and likely pathogenic variants in each DDR gene in each cohort. Two-sided Fisher exact tests were conducted in each gene in case and control cohorts. All reported *P* values were corrected to FDR using standard Benjamini–Hochberg procedure. FDR < 0.05

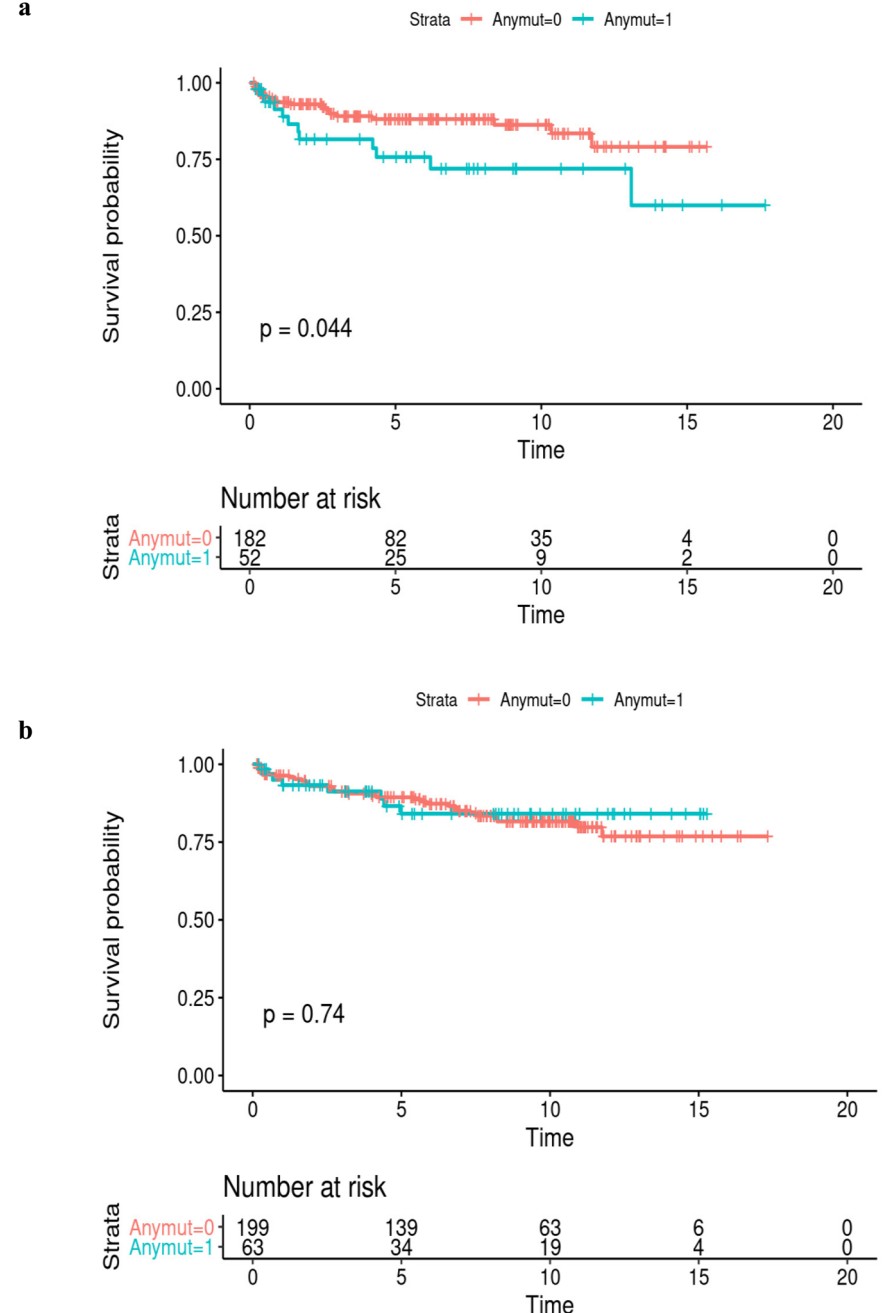

**Fig. 5 Kaplan–Meier Estimation Curve of Time to BCR as a Function of DDRG Germline Mutation Status.** DDRG mutation status (0, No mutation vs. 1, Any mutation), for CaP patients in AA (*N* = 259) (top figure, **a**) and European Ancestry patients (*N* = 272) (bottom figure, (**b**). The log-rank *p* value (*p* = 0.044) indicates that there is an association between DDRG germline mutation and BCR over 20 years of clinical follow up after radical prostatectomy in AA men. Source data are provided as a Source Data file.

was considered significant. Similar tests were also performed using all rare protein-altering SNP. Significant results were manually reviewed using the Integrated Genome Viewer to confirm the absence of alignment and possible technical artifacts.

**Pathogenic variant validation with droplet digital PCR (ddPCR).** The mutations identified by WGS were further confirmed by Droplet Digital Polymerase Chain Reaction (ddPCR) technique using a QX200 Droplet Generator (BioRad) and the data was analyzed by QuantaSoft software (BioRad). Briefly, a ddPCR mastermix was prepared containing 11 μl 2X ddPCR Supermix (BioRad), 1.1 μl 20X TaqMan SNP Genotyping Assay (BioRad, ThermoFisher Scientific; Supplementary Table 6), and 7.9 μl nuclease-free water (Qiagen) per sample. The mastermix was prepared at room temperature and 20 μl was added to 2 μl (5 ng) of each DNA sample. Samples were loaded into individual wells of DG8TM cartridges (BioRad), and droplets were generated using a QX200 Droplet Generator (BioRad). For each sample, 40 μl

of droplet mix was then transferred to a 96-well plate, and PCR was performed in a thermal cycler using the following cycling conditions: 95 °C × 10 min; 40 cycles of [94 °C × 30 s, 60 °C × 60 s]; 98 °C × 10 s; 40 C × 10 min. The BioRad QX200 Droplet Reader was then used to assess droplets as positive or negative based on fluorescence amplitude. The QuantaSoft software (BioRad) was used to analyze droplet data.

**Clinico-pathologic associations.** Association analyses of the germline mutations with Gleason grade, early age onset, BCR, and metastasis, were performed by chi-square test, Cox proportional hazard modeling, Kaplan–Meier plots. and log-rank tests. Chi-square testing was used to evaluate the associations of categorical clinico-pathological variables. The presence of DDRG germline mutations was examined in relationship to early age onset (<50 years old), Gleason grade at RP (3 + 3,3 + 4,4 + 3/8-10), BCR (yes/no), and metastasis (yes/ no). Cox proportional hazard modeling was used to assess the association for

time to BCR and time to metastasis. Kaplan–Meier (KM) plots were used to visualize the effect of the time to event analysis. Log-rank test was used to estimate the effects of variants on the outcomes.

**Reporting summary**. Further information on research design is available in the Nature Research Reporting Summary linked to this article.

## Data availability

The germline mutation summary statistics datasets that support the findings of this study are available in public dataset repository, Figshare[21]. Additional study related information is included in the Article, Supplementary Information or Source Data file. Source data are provided with this paper.

## Code availability

Code available upon request.

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

## Acknowledgements

The authors thank The American Genome Center (TAGC) for Whole Genome Sequencing including DNA QC, sequencing library preparation and QC, and sequencing on NovaSeq system. The contents of this publication are the sole responsibility of the author(s) and do not necessarily reflect the views, opinions or policies opinions of Uniformed Services University of the Health Sciences (USUHS), the Henry M. Jackson Foundation for the Advancement of Military Medicine, Inc., the Department of Defense (DoD) or the Departments of the Army, Navy, or Air Force. Mention of trade names, commercial products, or organizations does not imply endorsement by the U.S. Government.

## Author contributions

I.K., K.B., L.R., G.S., E. M-M., J. R., and C.A. performed the experiments; G.P., S.S., I.K., and J.C provided the study concept and design; M.W., C.D., X.Z., D.N., G.P., I.K. J.C. and Y.C. were involved with data acquisition, analysis, or interpretation; paper was drafted by I.K., G.P., S.S., M.W., X.Z; paper edited by S-H.T., A.D., G.C., H.P., and C.T.; statistical analysis was provided by D.N., J.C., M.W., X.Z., and Y.C. Funding was obtained by S.S. (CDMRPL-16-0-DM160510), H.P. (IAA-A-HL-14-007), and G.C. (Center for Prostate Disease Research, Uniformed Services University Grant HU0001-20-2-0032); I.L.R., I.A.S., L.R., A.A., and D.Y. provided biospecimen and technical support. All authors discussed the results and commented on the paper.

## Competing interests

S.S. and G.P. are co-inventors on the pending patent: Mutations of All DNA Repair Genes in African American and Caucasian American Prostate Cancer Patients, PCT/US21/21136 (Patent Cooperation Treaty). The remaining authors declare no competing interests.
