## [Peer Review File · Nature Communications]

Germline mutation landscape of DNA damage repair genes in African Americans with prostate cancer highlights potentially targetable *RAD* genesREVIEWER COMMENTS

Reviewer #1 (Remarks to the Author): Expert in prostate cancer genetics, pharmacogenomics, and health disparities

The authors' study objective was to profile germline mutations in all known DDRGs (N =276) using whole genome sequences from blood DNA of a matched cohort of patients (300 African Americans and 300 Caucasian Americans) with primary prostate cancer to determine if knowledge of DDRG mutation status can enhance patient stratification for specific targeted therapies. The authors found only 13 pathogenic or likely pathogenic DDRG variants shared among AA and CA patients. Additionally, RAD1, RAD54L, RAD54B, PMS2 and BRCA genes were more frequently mutated in AA patients compared to CA patients.

While the genomic analysis methods implemented in this paper are standard (not novel), I think the results of this paper would be of high interest to investigators who study prostate cancer. Particularly, prostate cancer in men of African descent where significant health disparities in diagnosis, treatment and research currently exists.

The authors state that the results of their study could enhance patient stratification for specific targeted therapies. However, I don't find convincing evidence of this claim in my review of this manuscript. The authors could have spent more effort taking the significant gene variant findings and conducted other secondary analyses (e.g. pathway analysis) in order to gain a more informed sense of known and novel specific therapeutic targets that could be studied in the future by this research group or others who research prostate cancer in health disparate populations, most notably in African American men.

I do feel that this paper will influence thinking in the field because of its focus on a health disparate population, African Americans with prostate cancer.

The following are my concerns and questions about this manuscript:

Lines 143-148: How did the authors conclude that a test using this 8-gene panel would detect 11.9% (31 of 259) of AA CaP patients and only 5.8% (16 of 272) of CA CaP patients with potentially targetable mutations (P = 0.0017)? I don't recall reading how this was determined in my review of the manuscript so I think that additional clarification is necessary.

For Supplementary Figure 1. Principle Component Analysis (PCA) of Ancestry using the PEDDY program. Although I agree with the populations included, why was the 1K Genomes reference population "ASW" population not included in the principle component analysis of ancestry? The ASW reference cohort is comprised of African Americans from the United States Southwest. This cohort would likely have the same percentage of European and African admixture as the African Americans in your study. I think the PCA analysis should be repeated after also including the 1K Genomes ASW reference population.

Lines 97-98: Of the 11.5% (69/600) excluded from the analysis, how many were due to of the patients due to various sequencing QC criteria and how many were excluded due to a mismatch between genomic ancestry and self-reported race? Could more African American subjects in particular have been included if the ASW cohort had been included in the PCA analysis?

Lines 136-138: Why all significant RAD mutations were grouped together? The rationale for this was unclear to me. Please provide additional clarification/justification for doing this.

The statistical analyses appear to be appropriate and valid. I was encouraged to learn that the authors used the Benjamini-Hochberg procedure to control for FDR. With the exception of the comments made in the manuscript about patient stratification for specific targeted therapies, I think that a different researcher could reproduce the work described in this manuscript given the level of detail provided.

Overall, the attention to the differences in DDRG genes involved in hereditary prostate cancer

between AA and CA populations is noteworthy as it helps to address the disparities that exist for AA afflicted with prostate cancer who despite having the highest morbidity and mortality of the disease, are not well-represented in prostate cancer genomics research. Thank you for the opportunity to review this submission.

Reviewer #2 (Remarks to the Author): Expert in cancer genetics and epidemiology, biomarkers, and ancestry

Overview

The present study seeks to characterize the mutational landscape of DNA-damage repair genes (DDRGs) in a matched retrospective cohort study of Black and White men treated in an equal access healthcare setting. The authors demonstrate that 46 of 276 known DDRGs were mutated among study participants. Of the 46, there was mutational overlap for only 13 of the 46 genes between Black and White men.

Novel Findings

This study makes a significant contribution to the literature by using a whole-genome sequencing platform to identify mutations in the 276 DDRGs. Black men were more likely to harbor mutations with potentially therapeutic targets. This finding may be clinically important as currently available genetic testing platforms do not capture the full spectrum of DDRG mutations.

Interpretation

The authors may wish to consider reframing their discussion of the health disparity. The inequity present is that Black men are an understudied population. This study addresses that inequity and identifies additional mutations that may have clinical relevance that have not been previously described. The genetic variation in DDRGs may help to explain the observed differences between Black and White men as it pertains to prostate cancer incidence and outcomes. However, the observed genetic variation between races alone does not indicate a disparity.

The authors note that a limitation of current genetic testing is that prognostic variants were identified among cohorts of men who were predominantly White and who had advanced stage disease. While the current study provides greater breadth of the mutational landscape of DDRGs, DDRG mutations are most likely to occur at advanced disease stage. A more informative approach for a future study may include characterizing DDRG variants among Black men with metastatic/advanced stage disease.

Reviewer #3 (Remarks to the Author): Expert in prostate cancer genomics

The authors present a paper on the frequency of germline mutations in African American vs Caucasian American prostate cancer patients. Indeed the paper is well written, concise and of interest to the community.

My comments:

1) The main assumption that is used here concerns the categorization of mutations as pathogenic/likely pathogenic. The use of Clinvar and other databases that are used by geneticists is an acceptable way to circumvent this problem, however it would be stronger if indeed the authors would present evidence that the mutation they find are indeed pathogenic. This could be done by providing family histories of cases or analysis of actual cancer genomes for mutations in the same genes (second hit).

2) The main strength of the paper is that the authors represent in an improved manner (WGS, which is by definition comprehensive) the frequency of germline alterations in DDRG genes. It would help the impact of the paper if a table could be added where the results of this analysis and

the previous analyses would be summarized: showing which genes were missed in previous analyses and how the frequencies of the genes that concur differ in the different datasets.

3) finally the outcome analysis is somewhat flawed as it is retrospective and could suffer from a number of biases. Here again validation would be great. There have been cohorts of local prostate cancer that have been characterized using WGS, interrogating those with the question whether these mutations indeed confer a poor prognosis would be helpful.

Responses to Reviewer's Comments:

Reviewer #1

The authors' study objective was to profile germline mutations in all known DDRGs (N =276) using whole genome sequences from blood DNA of a matched cohort of patients (300 African Americans and 300 Caucasian Americans) with primary prostate cancer to determine if knowledge of DDRG mutation status can enhance patient stratification for specific targeted therapies. The authors found only 13 pathogenic or likely pathogenic DDRG variants shared among AA and CA patients. Additionally, RAD1, RAD54L, RAD54B, PMS2 and BRCA genes were more frequently mutated in AA patients compared to CA patients.

While the genomic analysis methods implemented in this paper are standard (not novel), I think the results of this paper would be of high interest to investigators who study prostate cancer. Particularly, prostate cancer in men of African descent where significant health disparities in diagnosis, treatment and research currently exists.

The authors state that the results of their study could enhance patient stratification for specific targeted therapies. However, I don't find convincing evidence of this claim in my review of this manuscript. The authors could have spent more effort taking the significant gene variant findings and conducted other secondary analyses (e.g. pathway analysis) in order to gain a more informed sense of known and novel specific therapeutic targets that could be studied in the future by this research group or others who research prostate cancer in health disparate populations, most notably in African American men.

Response: We thank the reviewer for the kind feedback and the important points raised. Based on the reviewer's suggestions, we have incorporated information on significant gene findings, pathway analysis and added relevant references in the manuscript (**pages #7 and #8**).

Considering significant gene findings in relation to clinical utility, we found that several *RAD* genes (*RAD51*, *RAD54L*, *RAD54B*), as well as *PMS2* and *BRCA1*, were among the most frequently mutated DDRGs in AA patients, but not in CA patients. These genes are part of targetable DDRG pathway, specifically the homologous recombination (HR) pathway, suggesting potential benefit for AA men.

As a secondary analysis suggested by the reviewer, we performed pathway analysis based on published DNA Damage Repair pathways (Knijnenburg *et al*, Cell report 2018). The analysis showed that out of known 14 DDR pathways, the HR pathway (with 15 mutations) is associated with disease progression to BCR (p 0.018), while the NHEJ pathway (with 3 mutations) is associated with metastasis (p 0.045). We have added these new results to the **new Figure 3; referred to on page #8, Line 157**) and the **new Supplementary Table 5 (referred to on page #8, line 158)**

Following the reviewer's advice, in an additional secondary analysis, we also functionally scored and validated the germline mutations using an *in silico* tool, enGenome. These new results, which further supports our findings, has been incorporated in **page #7 (line 134-137), page #13**

and page # 14 (290-292), and new Supplementary Table 3 (More details are provided under Reviewer 3 Question 1 Response).

I do feel that this paper will influence thinking in the field because of its focus on a health disparate population, African Americans with prostate cancer.

Response: We thank the reviewer for this comment.

The following are my concerns and questions about this manuscript:

1. Lines 143-148: How did the authors conclude that a test using this 8-gene panel would detect 11.9% (31 of 259) of AA CaP patients and only 5.8% (16 of 272) of CA CaP patients with potentially targetable mutations ($P = 0.0017$)? I don't recall reading how this was determined in my review of the manuscript, so I think that additional clarification is necessary.

Response: Based on the reviewer's question, we have added clarifications in the manuscript. We know from the sequence and ddPCR validation that 11.9% of the AA CaP patients had germline mutation in any of the 8 genes. Therefore, we assume that a test for these 8 genes will detect the same 11.9% of the cases with germline mutation. The same logic applies for the CA patients. Supplementary Table 6 provides the summary of these 8 candidate genes, which were selected based on their potential clinical utility (being part of potentially targetable DDR pathways), and mutation carrier frequency of over 1%. This information with relevant references has been included in **page #8 (lines 165-166)**.

2. For Supplementary Figure 1. Principle Component Analysis (PCA) of Ancestry using the PEDDY program. Although I agree with the populations included, why was the 1K Genomes reference population "ASW" population not included in the principle component analysis of ancestry? The ASW reference cohort is comprised of African Americans from the United States Southwest. This cohort would likely have the same percentage of European and African admixture as the African Americans in your study. I think the PCA analysis should be repeated after also including the 1K Genomes ASW reference population.

Response: We thank the reviewer for the comment on inclusion of ASW in PCA ancestry analysis. The ASW subpopulation group (N=61) within the AFR major population was incorporated in the ancestry analysis but was not specifically indicated. To address this point, we have now expanded the description of the PCA ancestry analysis by better defining the major populations from the 1000 Genome project that was used in our analysis, in the **Supplemental Figure 1 legend (Page 1, lines 6-12 of Supplementary Information)**.

3. Lines 97-98: Of the 11.5% (69/600) excluded from the analysis, how many were due to of the patients due to various sequencing QC criteria and how many were excluded due to a mismatch between genomic ancestry and self-reported race? Could more African American subjects in particular have been included if the ASW cohort had been included in the PCA analysis?

Response: Thank you for this comment. We note that we had already included ASW in the PCA analysis as described in point #2 above, so our method of genomic-ancestry inference is not an

explanation for sample exclusion. To further respond to this point, we have added detail on the reason for sample exclusion, on **page 5 (lines 95-97)**. Specifically, 26 samples were excluded because of low quality sequencing results due to DNA yields, fragment size and contamination, and 33 samples were further excluded considering differences in self-reported and genomically-inferred ancestry. Since the aim of our study is to compare AA and CA patients, we decided to exclude samples with a lack of agreement between self-reported race and genomically-inferred ancestry. This new information is now added to **page 5 on lines 95-97**.

4. Lines 136-138: Why all significant RAD mutations were grouped together? The rationale for this was unclear to me. Please provide additional clarification/justification for doing this.

Response: Based on the reviewer's suggestions, we have provided additional justifications in the manuscript (**page # 7**), on **lines 152-154**. In brief, one of the major findings from our study is that several *RAD* family genes (*RAD51*, *RAD54L*, *RAD54B*), in addition to *PMS2* and *BRCA1*, were among the most frequently mutated DDRGs in AA patients, but not in CA patients, when compared to the relevant control datasets (please see Figure 2). *RAD51* was found to be the most frequently mutated HR pathway gene in AA cohort (please see **new Figure 3**). Because *RAD* genes are functionally related, we grouped all *RAD* mutations together. We observed a greater mutation rate in AA (6.95%) than in CA patients (1.10%) ($P = 0.001$; OR 6.68) (Please see Supplementary Table 4). Additionally, literature supporting potential clinical utility of the mentioned *RAD* genes has been included in **page #7 (lines 144, 145, 149, 150, 152)**.

5. The statistical analyses appear to be appropriate and valid. I was encouraged to learn that the authors used the Benjamini-Hochberg procedure to control for FDR. With the exception of the comments made in the manuscript about patient stratification for specific targeted therapies, I think that a different researcher could reproduce the work described in this manuscript given the level of detail provided.

Response: We thank the reviewer for this comment regarding the methodological rigor in our manuscript. We have added detailed description and new references about the targeted therapies on **page #7 (lines 144, 145, 149, 150, 152) and page #9 (line 191-192)**

Reviewer #2 (Remarks to the Author): Expert in cancer genetics and epidemiology, biomarkers, and ancestry

1. The authors may wish to consider reframing their discussion of the health disparity. The inequity present is that Black men are an understudied population. This study addresses that inequity and identifies additional mutations that may have clinical relevance that have not been previously described. The genetic variation in DDRGs may help to explain the observed differences between Black and White men as it pertains to prostate cancer incidence and outcomes. However, the observed genetic variation between races alone does not indicate a disparity.

Response: We agree and thank the reviewer for pointing out the correct usage of the racial disparity term. We have corrected the manuscript text by using the term "racial differences" for "racial disparity" [**page# 3 (line 50), #4 (line 67) , #6 (lines 118, 119) and #9 (line 200)**]. Some

of the racial differences in DDRG germline mutations may eventually contribute to racial disparity if linked to disparate CaP incidence or outcome between AA and CA patients. Relevant references on racial disparity aspects of CaP are now included in **page# 4 (lines 81, 82)**.

2. The authors note that a limitation of current genetic testing is that prognostic variants were identified among cohorts of men who were predominantly White and who had advanced stage disease. While the current study provides greater breadth of the mutational landscape of DDRGs, DDRG mutations are most likely to occur at advanced disease stage. A more informative approach for a future study may include characterizing DDRG variants among Black men with metastatic/advanced stage disease.

Response: We thank the reviewer for making this important point. We have now added text for possible future studies in advanced CaP in AA men in the Discussion section of the manuscript (**Page# 11, lines 238-239**). Towards this future goal, we plan to collaborate with researchers who have access to specimens from advanced/metastatic CaP, especially in the context of AA men.

Reviewer #3 (Remarks to the Author): Expert in prostate cancer genomics

The authors present a paper on the frequency of germline mutations in African American vs Caucasian American prostate cancer patients. Indeed, the paper is well written, concise and of interest to the community.

My comments:

1) The main assumption that is used here concerns the categorization of mutations as pathogenic/likely pathogenic. The use of Clinvar and other databases that are used by geneticists is an acceptable way to circumvent this problem, however it would be stronger if indeed the authors would present evidence that the mutation, they find are indeed pathogenic. This could be done by providing family histories of cases or analysis of actual cancer genomes for mutations in the same genes (second hit).

Response: We agree with the reviewer that providing additional evidence on variant functionality (P/LP) would strengthen our ClinVar based P/LP classification. As per the reviewer's suggestions, we have functionally scored and validated the germline mutations using the enGenome *in silico* tool based on ACMG, AMP, ClinGen guidelines. These new results support our prior ClinVar-based results and are now incorporated in **pages #7 (lines 134-137), #13 and #14 (lines 290-292)** and **new Supplementary Table 3**. In brief, the table shows that 98/100 variants were functionally validated by enGenome prediction tool [Expert Variant Interpreter (eVai) version 2], which provides a Pathogenicity score computed based on ACMG/AMP guidelines and corresponding levels of evidence using various prediction tools (PaPI, DANN, dbScSNV, PolyPhen, SIFT).

Unfortunately, family cancer history data is largely unavailable in our clinical database, so we are unable to perform a family history analysis. The reviewer also states that somatic genomes sequencing could be incorporated into the P/LP germline variant interpretation for individual cases. This is an excellent suggestion, and we are aspiring to address somatic genome sequencing in CaP tumors in a future study.

2) The main strength of the paper is that the authors represent in an improved manner (WGS, which is by definition comprehensive) the frequency of germline alterations in DDRG genes. It would help the impact of the paper if a table could be added where the results of this analysis and the previous analyses would be summarized: showing which genes were missed in previous analyses and how the frequencies of the genes that concur differ in the different datasets.

Response: We thank the reviewer for the suggestion. As a response to this suggestion, we compared the results of our current study to 18 similar published studies. We have now added these results to a new table **Supplementary table 1 (referred to on Page#6, lines 128-130 and Page#10 and #11 (lines 216-217, 220-224))**, comparing candidate genes, gene frequency, sequencing strategy, cancer stage and cohort characteristics between the present study and the others. In general, our rates of DDRG mutations were in the range of prior studies in prostate cancer; however, our study contains the first report of mutations in several DDRG, especially in AA cases (such as in HR genes). *RAD51* plays a crucial role in finding the homologous DNA strand during HR. Our study detected germline mutations in this gene exclusively in AA patients with a 2.7% frequency. To our knowledge, no other study detected germline mutations in *RAD51* in prostate cancer context (it was not mutated in CA cases in our study either). Similarly, we uniquely report mutations in *RAD54B*, also completely specific to AA cases (with 1.5% frequency), but not in CA cases. We detected germline mutations in *RAD54L* in AA cases (with 1.5% frequency) and also in CA cases (with 0.7% frequency). One of these 18 papers (Castro *et al*, 2019) also detected germline mutations in *RAD54L* (with 0.2% frequency). Finally, in our study the number of DDRGs tested for was 276 (all known DDRGs), others specifically tested for a selected subset of 2-177 DDRGs. Consequently, the number of DDRGs with germline mutations detected was 46 in our study, versus 2-29 in the other papers.

3) Finally, the outcome analysis is somewhat flawed as it is retrospective and could suffer from a number of biases. Here again validation would be great. There have been cohorts of local prostate cancer that have been characterized using WGS, interrogating those with the question whether these mutations indeed confer a poor prognosis would be helpful.

Response: We agree with the reviewer that outcome analysis may have bias considering retrospective nature of the study, however this study was performed in a blinded design (prospective-specimen-collection, retrospective-blinded-evaluation (PRoBE) design (Pepe *et al*, JNCI, 2008).

The reviewer also indicates that validation would enhance our outcome analysis. We interpret this point as a request for replication of the DDRG mutation vs survival outcome in an independent cohort. We need to point out that our current cohort is highly unique as we have long clinical follow up time for both AA and CA patients in a Military Health System with equal access to healthcare. There is no publicly available cohort with these 5 characteristics (WGS/WES, AA patients, CA patients, equal access to healthcare, and 10-year BCR follow up time), so the exact validation from the published studies is not feasible considering cohort characteristics and clinical follow up time.

Most studies in the literature using WGS approach in primary (localized) prostate cancer are focused on CA cohorts. However, a recently published study (Matejcic *et al*, JCO Precis Oncol.

2020) on 19 DDRGs in 2,453 African American and 1,151 Ugandan cases and controls with prostate cancer corroborates our result on the association of DDRG germline mutations with aggressive CaP (**Supplementary Table 7**). They found that highest risks for aggressive disease were observed with pathogenic variants in *ATM*, *BRCA2*, *PALB2*, and *NBN* genes. In our analysis, we also found that *BRCA2* and *NBN* are 2 of 5 genes (*RAD51*, *BRCA1*, *BRCA2*, *BLM*, *NBN*), where germline mutations are enriched (in HR pathway) and associates with BCR. Therefore, we have added this new validation to our discussion on **Page #10 and # 11 (line 220-224)** and mentioned the need for future validation study in the manuscript on **Page #11, lines (229-231)**.

We have also compared our study with other similar studies in **Supplementary Table 1** and in the **Discussion section** of the manuscript on pages **#10 and #11** (More details are provided under Reviewer 3 Question 2 Response).

REVIEWERS' COMMENTS

Reviewer #1 (Remarks to the Author):

Thank you for your responses to my review of this manuscript.

Reviewer #2 (Remarks to the Author):

Thank you for your revision. The response and revised manuscript adequately address the reviewer's concerns.

Reviewer #3 (Remarks to the Author):

the concerns were well addressed